# Effects of GPR18 Ligands on Body Weight and Metabolic Parameters in a Female Rat Model of Excessive Eating

**DOI:** 10.3390/ph14030270

**Published:** 2021-03-16

**Authors:** Magdalena Kotańska, Kamil Mika, Małgorzata Szafarz, Monika Kubacka, Christa E. Müller, Jacek Sapa, Katarzyna Kieć-Kononowicz

**Affiliations:** 1Department of Pharmacological Screening, Faculty of Pharmacy, Jagiellonian University, Medical College, 9 Medyczna Street, 30-688 Kraków, Poland; kamil.mika@doctoral.uj.edu.pl (K.M.); monika.kubacka@uj.edu.pl (M.K.); jacek.sapa@uj.edu.pl (J.S.); 2Department of Pharmacokinetics and Physical Pharmacy, Jagiellonian University Medical College, 9 Medyczna Street, 30-688, Krakow, Poland; malgorzata.szafarz@uj.edu.pl; 3Pharmaceutical & Medicinal Chemistry, Pharmaceutical Institute, PharmaCenter Bonn, University of Bonn, An der Immenburg 4, D-53121 Bonn, Germany; christa.mueller@uni-bonn.de; 4Department of Technology and Biotechnology of Drugs, Faculty of Pharmacy, Jagiellonian University, Medical College, 9 Medyczna Street, 30-688 Kraków, Poland; katarzyna.kiec-kononowicz@uj.edu.pl

**Keywords:** GPR18 ligands, anorectic activity, palatable diet, excessive eating model, PSB-CB5

## Abstract

GPR18 has been proposed to play a role in the progression of metabolic disease and obesity. Therefore, the aim of this study was to determine the effects of selective GRP18 ligands (the antagonists PSB-CB5 and PSB-CB27 and the agonist PSB-KK1415) on body mass and the development of metabolic disorders commonly accompanying obesity. Experiments were carried out on female Wistar rats. In order to determine the anorectic activity of the investigated ligands, their effect on food and water intake in a model of excessive eating was assessed. Lipid profile, glucose and insulin levels as well as alanine aminotransferase, aspartate aminotransferase, and γ-glutamyl transpeptidase activity in plasma were also evaluated. Potential side effects were examined in rat models of pica behavior and conditioned taste aversion. Animals treated with different ligands gained significantly less weight than rats from the obese control group. Effects of GPR18 antagonists on food intake and body weight were specific and unrelated to visceral illness, stress or changes in spontaneous activity. However, the GPR18 agonist is likely to affect body weight by inducing gastrointestinal disorders such as nausea. The presented preliminary data support the idea that the search for selective GPR18 antagonists for the treatment of obesity might be promising.

## 1. Introduction

The endogenous cannabinoid system consists of substances acting as neuromodulators i.e., endocannabinoids, their receptors and enzymes that are responsible for their biosynthesis and degradation [1]. Endocannabinoids bind to several receptors, including CB_1_, CB_2_, GPR18, GPR55, and GPR119, peroxisome proliferator-activated receptors and transient receptor potential vanilloid type 1 (TRPV1) [2,3,4]. They are synthesized from arachidonic acid and other polyunsaturated fatty acids. Anandamide (AEA) and 2-arachidonoylglycerol (2-AG) were the first endocannabinoids discovered; they are most abundant in the human brain [5]. AEA partially, whereas 2-AG fully activate both CB_1_ and CB_2_ cannabinoid receptors [6,7]. Other endocannabinoids include 2-arachidonoylglyceryl ether (noladin ether), O-arachidonoylethanolamine (virodhamine), and *N*-arachidonoyl-dopamine [8].

GPR18, identified in 1997, has been reported to be activated by Δ^9^-tetrahydrocannabinol (THC), AEA [9], and *N*-arachidonylglycine—the endogenous metabolite of AEA [10] and resolvin D2 [11]. However, there are contradicting reports on several proposed putative GPR18 agonists [12,13,14,15], and the only one which is generally confirmed is THC; therefore, GPR18 has been considered as a third cannabinoid receptor [9]. GPR55 is proposed to be activated endogenously by lysophosphatidylinositol [16]. GPR119 is activated by monounsaturated fatty acid analogues of AEA and 2-AG, *N*-oleoylethanolamine and 2-oleoylglycerol [17], but does not appear to respond to plant-derived or synthetic cannabinoids. GPR18 and GPR55, however, have been suggested to be targets for some of these agents [2].

Endocannabinoids are lipid messengers involved in overall body weight control by interfering with multiple central and peripheral regulatory mechanisms that coordinate energy homeostasis [18]. They influence the energy metabolism through central modulation of caloric intake, as well as peripheral changes in nutrient transport, cellular metabolism, and energy storage [19]. However, in order to treat overeating, even though pharmacological blockade of cannabinoid receptor type 1 (CB1) by its inverse agonist rimonabant admittedly suppressed feeding it also resulted in psychiatric side effects. Therefore, research within the last decade focused on deciphering the underlying cellular and molecular mechanisms of central cannabinoid signaling that control feeding as well as other behaviors, with the overall aim being the identification of specific targets to develop safe pharmacological interventions for the obesity treatment [20].

Since some researchers suggest the possible role of GPR18 in the progression of metabolic disease and obesity, this receptor and its ligands became a new potential therapeutic target [10]. So far, only few (mostly nonselective) GPR18 agonists [15] or antagonists have been described [21,22,23]. Our groups have recently developed potent and selective GPR18 agonists, e.g., compound PSB-KK1415. Additionally, several selective antagonists of GRP18 were obtained [22,23]—e.g., PSB-CB27 and PSB-CB5—the last one being the first potent and strongly preferential GPR18 antagonist, now commercially available (product name CID-85469571) from several companies. Since previously known GPR18 antagonists also antagonize GPR55 [24], it has been difficult to separate the effects of these two receptor targets. Due to the recent availability of selective GPR18 ligands these compounds may become useful tools in research focused on the mechanisms of action of this receptor. The aim of our work was to determine the effects of the above-mentioned GRP18 ligands (PSB-CB5, PSB-CB27, and PSB-KK1415) on body mass and selected metabolic disorders commonly accompanying obesity, in a female rat model of excessive eating. The contribution of the receptor itself on the development of obesity and hyperglycemia, as well as its influence on the lipid profile have also been taken into account. The research project was carried out using a rat model of excessive eating which perfectly illustrates the excess caloric intake from the excessively available tasty products rich in sugar and fat. In this model, the animals have access not only to high-calorie foods such as peanuts, cheese, milk with increased fat content, and chocolate, but also to the standard feed. However, feeding is not in any way forced, as in other models where animals are fed only a high-fat diet, or when they are temporarily deprived of food (binge eating models). The experience related to this model suggests that greater differences (requiring less animals per group in order to obtain significant results) in the severity of disturbances in body weight and selected metabolic parameters are observed in female rats [25,26,27]. Therefore, to make the effect more noticeable, and in line with the principle of reducing the number of animals, female rats were selected. 

## 2. Results

### 2.1. Effect of Diet or Tested Compounds on Body Weight, Amount of Intraperitoneal Adipose Tissue, Caloric, and Water Intakes

Animals fed a palatable diet gained more weight than rats from the control group, starting from the second day of the experiment. Throughout the experiment (21 days), control rats gained only 18.32% weight, while the rats from the control obese group gained 36.40% (Figure 1A). At the end of experiment, animals fed a palatable feed and treated with all tested compounds, namely PSB-KK1415 (GPR18 agonist), PSB-CB27, or PSB-CB5 (GPR18 antagonists), or rimonabant (CB1 antagonist), weighted significantly less than animals from the obese control group. However, the rats’ body weight in all these groups was still significantly higher than in the control group fed with a standard feed (Figure 1A,B). 

### 2.2. Effect of Diet or Tested Compounds on the Weight of Selected Organs 

Animals from the control obese group had a statistically higher amount of peritoneal fat pads (about 50% more) than control rats (Figure 2A). However, animals from all treated groups had significantly less peritoneal fat pads than control obese rats (Figure 2A). Additionally, the heart weight was lower in rats treated with all tested compounds compared to the control group fed a standard feed, and lower in the groups receiving the antagonists as compared to the control group fed palatable feed (Figure 2B). The kidney mass was significantly higher in the obese control animals and rats treated with PSB-KK1415 vs. control animals (Figure 2C). There were no statistical differences between the studied groups in the liver weight (Figure 2D).

### 2.3. Effect of Diet or Tested Compounds on Caloric and Water Intakes

Calorie intake in all groups was significantly higher than in the control group (Figure 3A), and in the case of PSB-KK1415 treatment the difference was also significant vs. the control obese rats. Significantly less water was consumed by obese control animals, which could probably be explained by the fact that they had also access to milk. However, the rats fed palatable feed and treated with PSB-KK1415, PSB-CB5, or rimonabant drank significantly more water than obese control animals. Only animals treated with PSB-CB27 consumed a similar amount of water as control animals fed a preferential feed (Figure 3B).

### 2.4. Influence of Diet or Tested Compounds on Plasma Glucose and Insulin Levels

All animals fed palatable feed had significantly higher blood glucose levels than control rats. Additionally, animals treated with PSB-KK1415 had blood glucose levels even higher than rats from the obese control group (Figure 4A). An agonist of GPR18—PSB-KK1415, one of antagonists—PSB-CB5, and rimonabant increased levels of insulin in plasma of rats fed a palatable feed, comparing to the level of insulin in control obese animals. However, only the increase observed in PSB-KK1415 treated group was statistically significant. Treatment with another GPR18 antagonist, compound PSB-CB27, did not influence the insulin level which stayed similar to the one observed in plasma of control obese rats (Figure 4B).

### 2.5. Influence of Diet or Tested Compounds on Lipid Profile

The level of triglyceride in blood was higher in obese rats than in the control group, but the HDL-cholesterol level was lower (Figure 4C,E). Rats treated for three weeks with rimonabant or test compounds, except PSB-KK1415, and fed palatable diet had slightly lower levels of triglycerides in plasma than animals from the obese control group. There was no statistical significance between levels of triglycerides in these groups and the control group. However, in the PSB-KK1415-treated group the triglyceride level was significantly higher than in the control group (Figure 4C). There were no significant differences in total cholesterol plasma levels between all studied groups (Figure 4D). Groups fed a preferential feed and treated with all test compounds had a higher level of HDL-cholesterol than the obese control rats, and in the cases of PSB-KK1415 or rimonabant administration it was even higher than in the control animals (Figure 4E).

### 2.6. Influence of Diet or Tested Compounds on Alanine Aminotransferase (AlAT), Aspartate Aminotransferase (AspAT), and γ-Glutamyl Transpeptidase (GGT) Activity in Rats Plasma

A significantly higher level of AlAT activity was observed in plasma of rats treated with PSB-KK1415 or PSB-CB5 compared to the levels in both control groups (Figure 4F). A significantly higher level of AspAT activity was observed in plasma of rats treated with PSB-KK1415 or PSB-CB5 compared to the level in control groups fed palatable feed (Figure 4G). The GGT activity was significantly lower in plasma of rats treated with PSB-KK1415 compared to the levels in the control group fed palatable feed (Figure 4H).

### 2.7. Influence of Single and Chronic Administration of the Tested Compounds on Locomotor Activity of Rats Fed Palatable Diet and Housed in Pairs in Home Cages 

All tested compounds at the administered doses had no effect on locomotor activity after both single and chronic administration to rats fed preferential feed (Figure 5).

### 2.8. Effects on Visceral Illness via Measurement of Kaolin Intake (Pica Behavior Model)

Significantly lower weight gain over the 24-h period of the study was observed in the groups treated with PSB-KK1415 or PSB-CB27 vs. control group treated with 1% Tween 80 (negative control). The positive control group was given CuSO_4_ and was characterized by the loss of weight, the greatest intake of kaolin and the smallest intake of feed (Figure 6A). Animals receiving PSB-KK1415 or rimonabant consumed significantly more kaolin than rats from the negative control group, however, less than rats from the positive control group. While, animals receiving PSB-CB27 or PSB-CB5 consumed kaolin in the amount comparable to the negative control group (Figure 6B). Animals from all groups receiving tested compounds ate significantly less than animals from control group that received only vehicle and significantly more than rats from group treated with CuSO_4_ (Figure 6C). There were no statistically significant differences in water intake between groups (Figure 6D). Animals from the groups treated with CuSO_4_ or PSB-CB27 excreted significantly less feces within 24 h after compound administration compared to the control group (Figure 6E).

### 2.9. Effects on Visceral Illness via Measurement of Saccharin Intake

PSB-CB5 and PSB-CB27 did not induce any significant conditioned taste aversion measured by saccharin preference following a seven-day conditioning period in which rats were administered (i.p.) above mentioned compounds. In contrast, PSB-KK1415, the reference compound LiCl (40 mg/kg b.w.) and rimonabant caused a significant reduction of saccharine consumption. Slight, however significantly lower weight gain over the 24-h period of the study was observed in the groups treated with PSB-KK1415, PSB-CB27, LiCl, or rimonabant (Figure 6F,G).

## 3. Discussion

Obesity may be, at least partially, linked to dysregulation of the endocannabinoid system in both central and peripheral tissues, affecting appetite as well as glucose and lipid metabolism [28,29,30]. Therefore, in order to determine the pharmacological activity of GPR18 ligands we have chosen the model of excessive overeating with preferential feed, in which animals have unrestricted access to the standard diet with addition of products such as peanuts, cheese, milk, and chocolate. The availability of such diet results in increased caloric intake and development of hyperglycemia, lipid disorders, and obesity with accompanying increase of the amount of peritoneal fat [25,26,31]. Indeed, in our study, within 21 days of being on such a diet rats were overweight and developed the above-mentioned metabolic disorders. By treating animals with the selective GPR18 ligands we aimed to determine if this receptor could be, to some extent, related to dysregulation of the endocannabinoid system in obesity. The particular aim of this study was to identify whether the administration of selective GPR18 ligands would prevent induction of at least some pathological disorders accompanying obesity. We also wanted to determine whether these compounds could act anorectically and which ligand—agonist or antagonist—would have the most beneficial effect. 

Endocannabinoids and non-selective GPR18 agonists, such as AEA or 2-arachidonoyl glycerol increase food intake and consequently body weight by activating central endocannabinoid receptors [28,32,33]. It was hypothesized [34] that the high efficacy of cannabinoid receptor inverse agonists such as rimonabant in reducing body weight of obese animals and humans (regardless of the food intake inhibition), is due to the occurrence of a general up-regulation of the endocannabinoid system in obesity, not only at the central [34], but also at the peripheral [32,35,36] levels. Additionally, over reactivity of the endocannabinoid system can increase accumulation of fat and reduce glucose uptake in human fat cells, subsequently increasing the risk of insulin resistance and impaired glucose tolerance [28]. These data let us believe that GPR18 antagonists could exert beneficial effects in reducing both, body weight and symptoms of metabolic disorder in obese animals. Surprisingly, animals treated for 21 days with different ligands (GPR18 agonist, antagonists and CB1 antagonist) gained significantly less weight than rats from the control obese group. However, this effect was observed only in the third week of treatment except for the reference compound (rimonabant), which significantly slowed down the animals’ weight gain already from the first week of administration. Additionally, in all these animals significantly smaller amount of peritoneal adipose tissue was observed than in the control obese rats.

After obtaining such results, it became obvious that we had to rule out other possible effects exerted by the tested compounds, such as sedation or gastrointestinal disorders, which could also contribute to the slowing down the weight gain and reduction of the amount of peritoneal fat in experimental animals. Stress as well as changes in the spontaneous activity could be very disadvantageous and distort an assessment of the impact of tested compounds on body weight [37]. Sickness, gastrointestinal malaise, or drug-induced toxicity can also be the source of weight loss causing changes in food consumption or acting through some other mechanisms (e.g., malabsorption) [38].

The effect of tested compounds on the spontaneous activity was investigated in the group of animals fed a palatable feed. In order to reduce stress, rats were housed in pairs in home cages. The activity monitoring system was imperceptible to the animals and did not affect their behavior in any way. None of the tested compounds had a statistically significant effect on rats’ mobility after both single and chronic administration. Thus, the changes in activity could be ruled out as the potential cause of reduced weight gain.

Some drugs may reduce food intake by producing gastrointestinal malaise, which is difficult to detect solely by the changes in animal behavior. Rats and mice lack the emetic response, which distinguishes them from humans. However, the persistent eating of inedible substances by rodents can be used to evaluate illness-response behavior analogous to vomiting in other species [38,39]. Thus, in our study, animals beside preferential feed had also access to kaolin clay. Administration of GPR18 antagonists PSB-CB27 or PSB-CB5 did not result in the excessive intake of clay (compared to the negative control group receiving vehicle only) which proved that in these animals’ disorders of the gastrointestinal tract, such as visceral irritation or nausea, did not occur. On the other hand, CuSO_4_, used as a positive control, extensively reduced the food and water intake and significantly increased the amount of consumed kaolin clay, indicating the stomach upset. Unfortunately, the administration of PSB-KK1415 (GPR18 agonist) caused analogous changes in animals’ behavior. Thus, in rats treated chronically with this compound, disorders of the gastrointestinal tract could be, at least partially, responsible for the reduced weight gain. After a single PSB-KK1415 administration rats consumed less food than ones from the negative control group, however, the amount of excreted feces in both groups was similar what may also suggest that PSB-KK1415 has a potential ability to increase the intestinal passage. 

Sucrose preference test is often used to examine possible side effects such a nausea or malaise. In most cases, this test is performed in acute settings. However, the most relevant question, mainly from a therapeutic point of view, is whether the support of potential conditioned tasted aversion (CTA) will be sustained during repeated administration [40]. The results of seven-day subchronic CTA test showed that compounds PSB-CB5 and PSB-CB27 did not cause gastrointestinal disturbances. In contrast, PSB-KK1415 (same as rimonabant) caused an adverse reaction of similar intensity to the one produced by the reference compound (lithium chloride), that clearly indicated development of the gastrointestinal disorders in these animals. This adds further support to the conclusion that effects of GPR18 antagonists PSB-CB5 or PSB-CB27 on food intake and body weight are specific and unrelated to visceral illness, stress, or changes in the spontaneous activity. In contrast, GPR18R agonist compound PSB-KK1415 is likely to affect body weight by inducing gastrointestinal disorders, such as nausea.

An appropriate insulin sensitivity and the activation of its downstream machinery is a fundamental prerequisite to allow adequate fuel supply to the fat cells [41], therefore, impaired glucose transport and insulin resistance are the source of fat storage disorders. Inversely, the excess of free fatty acids released by visceral adipose tissue is the cause of “lipotoxicity”. Increased fat oxidation in the muscles causes inhibition of glycolysis, and in the liver contributes to the intensification of gluconeogenesis, which requires compensatory secretion of insulin by β cells. In both cases, plasma glucose, triglyceride, and insulin levels are elevated. In the presented study such metabolic changes were observed in rats treated with PSB-KK1415 (GPR18 agonist). Animals receiving this compound, despite lower weight gain, consumed significantly more kcal even than control obese animals. The body’s response was only partially analogous to that of proposed endogenous GPR18 agonists—AEA or 2-arachidonoyl glycerol [28,32,33] and the proposed exogenous agonist—O-1602 [42] all of which caused both increase in the food intake and weight gain. However, it should be emphasized that the abovementioned compounds are not, or not only, targeting GPR18 in contrast to PSB-KK1415 that is a selective GPR18 agonist. Moreover, in the process of developing metabolic disorders such as diabetes, after the initial period of weight loss, usually gain of weight occurs, so it is possible that longer studies would show a similar trend after chronic PSB-KK1415 administration.

Animals treated with PSB-KK1415 also drank more water than rats from both control groups, despite the fact that animals eating the preferential feed also consumed milk. Taking into account that the typical symptoms of developing diabetes are polydipsia and weight loss, as well as elevated glucose levels, it can be concluded that the administration of PSB-KK1415 may cause the development of pre-diabetes or diabetes and insulin resistance. Previously, Ikeda et al. showed that 2-arachidonoyl glycerol causes increased insulin release from rat islet cells [43]. However, to confirm our hypothesis further studies, such as glucose tolerance and insulin sensitivity tests after PSB-KK1415 administration are needed. 

Fatty liver is usually associated with elevated levels of AlAT and AspAT in plasma [44]. Yet, studies suggest that AlAT levels can fluctuate, therefore, measurement at a single time point is considered to be insufficient to draw conclusions regarding the liver function [45]. In the present study, no high changes in the activity of all three tested liver enzymes (AlAT, AspAT, and GGT) were observed. While some changes were statistically significant, they could not be considered clinically alarming.

In conclusion, the present study has addressed a number of critical issues in the evaluation of the concept of using GPR18 ligands to decrease food intake and lower body weight gain. Animals treated with different ligands, both agonists and antagonists, gained significantly less weight than rats from the control group fed palatable feed. However, the effects of GPR18 antagonists on food intake and body weight were specific and unrelated to visceral illness, stress or changes in the spontaneous activity, while the GPR18 agonist was likely to affect body weight by inducing gastrointestinal disorders, such as nausea. Rats treated with GPR18 antagonists had also slightly lower levels of triglycerides in plasma than animals from the obese control group. Thus, the presented preliminary data support the idea that selective GPR18 antagonism represents a promising avenue for the further research, and that the development of selective GPR18 antagonists for the treatment of obesity might result in valuable and promising outcomes. Currently, the most urgent issue seems to be the identification of a potential candidates with better efficacy and superior pharmacological activity than current drugs towards the normalization of lipid and carbohydrate disorders as well as insulin resistance, a metabolic condition often accompanying obesity.

## 4. Materials and Methods

### 4.1. Animals and Tested Compounds

Experiments were carried out on female Wistar rats with the initial body weight in the range of 165–175 g (six weeks old: Jagiellonian University Medical College, Krakow, Poland). The animals were housed in plastic cages in constant temperature facilities exposed to 12–12 light-dark cycle. Water and food were available ad libitum. Control and experimental groups consisted of six to eight animals each. 

Tested compounds suspended in 1% Tween 80 were administered intraperitoneally (i.p.) once a day for 21 days at the dose of 5 mg/kg b.w. (PSB-KK1415; PSB-CB27; PSB-CB5) or 1 mg/kg b.w. (rimonabant—reference compound). The dose of tested compounds was chosen based on preliminary locomotor activity studies (unpublished observations). One non-sedative dose common to all compounds was selected. There were also two control groups, one fed a palatable diet (control obese group) and one fed a standard diet (control group). 

### 4.2. Drugs, Chemical Reagents, and Other Materials

Heparin was purchased from Polfa Warszawa S.A. (Warsaw, Poland), thiopental sodium from Sandoz International (Stryków, Poland), Tween 80 from Sigma-Aldrich (Darmstadt, Germany) and rimonabant was from AK Scientific, Inc., USA.

GPR18 ligands: agonist (PSB-KK1415 with undisclosed structure) and antagonists: (PSB-CB5-(Z)-2-(3-(4-chlorobenzyloxy)benzylidene)-6,7-dihydro-2*H*-imidazo [2,1-*b*][1,3]thiazin-3(5*H*)-one) and (PSB-CB-27-(Z)-2-(3-(6-(4-chlorophenoxy)hexyloxy)benzylidene)-6,7-dihydro-2*H*-imidazo[2,1-*b*][1,3]thiazin-3(5*H*)-one) were synthesized at the Department of Technology and Biotechnology of Drugs, Faculty of Pharmacy, Jagiellonian University Medical College, Krakow, Poland according to a procedure described previously [20,21]. Agonist: (PSB-KK-1415 hGPR18 − EC_50_ = 0.0191 (μM), β-arrestin recruitment assay; hGPR55 − EC_50_ > 10 (μM)), antagonist (PSB-CB5 − IC_50_ = 0.279 (μM) in β-arrestin recruitment assay as inhibitor of GPR18 activation by THC, hGPR55, IC_50_ > 10 (μM); hCB_1_, K_i_ > 10 (μM); hCB_2_, K_i_ = 4.03 (μM)) and antagonist (PSB-CB27 − IC_50_ = 0.650 (μM), full inhibition of THC induced activation of GPR18 in β-arrestin recruitment assay, hGPR55, IC_50_ > 10 (μM); hCB_1_, K_i_ > 10 (μM); hCB_2_, K_i_ > 10 (μM)). The compounds were discovered in the laboratory of C.E. Müller, University of Bonn, Germany.

### 4.3. Effect of Tested Ligands on Body Weight, Locomotor Activity, and Food and Water Intake

In order to determine the anorectic activity of tested ligands, its effect on food and water intake in the model of excessive eating was assessed [25,26]. Rats were housed in groups of three. Five groups of six rats were fed diet consisting of milk chocolate with nuts, cheese, salted peanuts, and 7% condensed milk and also had access to standard feed (Labofeed B, Morawski Manufacturer Feed, Poland) and water ad libitum for three weeks. All tested compounds were suspended in 1% Tween 80 and administered i.p. once a day (from the first day of experiment) for 21 days. First experimental group was given vehicle (1% Tween 80)—obese control group, second was given PSB-KK1415 (GPR18 agonist) at the dose of 5 mg/kg b.w., third group was given PSB-CB27 (GPR18 antagonist) at the dose of 5 mg/kg b.w., fourth group was given PSB-CB5 (GPR18 antagonist) also at the dose of 5 mg/kg b.w. and the fifth group was given rimonabant (CB1 antagonist) at the dose of 1 mg/kg b.w. The last group of rats was fed a standard feed for 3 weeks ad libitum and was given i.p. a vehicle (1% Tween 80)—control group. Intakes of food and water were evaluated three times per week, and body weights were measured daily immediately prior to the administration of the investigated compounds. 

Palatable diet contained: 100 g peanuts—614 kcal; 100 mL condensed milk—131 kcal; 100 g milk chocolate with hazelnuts—195 kcal; 100 g Greek cheese—270 kcal. The standard diet contained 100 g feed—280 kcal.

The locomotor activity of rats was measured on the first and 20th day of experiment with a TraffiCage (TSE-Systems, Germany) radio-frequency identification system (RFID) [46]. The animals had subcutaneously implanted transmitter identification, which enabled the presence and time spent in different areas of the cage to be recorded, and then data were collected with a special computer program. Locomotor activity was monitored for 18 h after single and repeated administration of the tested compounds.

On the 22nd day, 20 min after i.p. administration of heparin 600 U/rat, animals were sacrificed. Peritoneal fat, liver, kidneys, and heart were weighed, and plasma was collected for further analysis. Scheme of experiment is showed in Figure 7A.

### 4.4. Influence of Tested Compounds on Lipid Profile, Glucose and Insulin Levels as Well as AlAT, AspAT, and GGT Activity in Plasma

To determine the lipid profile, glucose level and the AlAT, AspAT, and GGT activity in plasma standard enzymatic and spectrophotometric tests (Biomaxima S.A. Lublin, Poland) were used. To determine the insulin level in plasma, an ELISA test (Fine Test, Wuhan Fine Biotech Co., Ltd., China) was used, performed in two replicates (standard curve range: 15–480 ng/mL (0.1 mIU/L–40 mIU/L), sensitivity: 0.05 mIU/L, intra-assay: CV < 8%, inter-assay: CV < 10%).

### 4.5. Effects on Visceral Illness via Measurement of Kaolin Intake (Pica Behaviour)

To exclude the possibility that the suppression of food intake by tested compounds was caused by visceral illness, Pica behavior was evaluated. The method was based on the works by Takeda [39], Yamamoto [47], and Kotańska [26]. The experiment lasted eight days. In addition to free access to feed, animals had also free access to the white kaolin. For the first few days animals were accustomed to the presence of kaolin in their cages. On the sixth day, food and kaolin were removed for 24 h, so the animals would be more or less at the same level of hunger and have a similar amount of feces in the digestive tract, since at the end of experiment feces were weighted to assess the level of intestinal passage. On the seventh day animals were i.p. given: tested compounds at a dose of 5 mg/kg b.w. or rimonabant at a dose of 1 mg/kg b.w. or vehicle (negative control group) or a solution of CuSO_4_ at a dose of 6 mg/kg b.w. (1/3 LD_50_; LD_50_ = 18 mg/kg for a rat at this route of administration) (positive control group). The standard feed and kaolin were given back to the cages. The amount of approved food, water drunk, kaolin consumed and excreted feces was determined after 24 h. Animals were also weighed prior to the administration of the tested compound and after 24 h. Scheme of the experiment is showed in Figure 7B.

### 4.6. Effects on Visceral Illness via Measurement of Sucrose Intake (Conditioned Tasted Aversion; CTA)

A seven-day subchronic CTA paradigm was adopted in order to examine whether tested compounds induced CTA following repeated administration. The method was based on the works by Malmlöf [40] with minor modification. The experiment lasted seven days. Animals were divided into six groups: one group received daily i.p. injections of vehicle (negative control group), a second group received a solution of lithium chloride (LiCl, 40 mg/kg b.w., i.p., positive control group), a third group was given rimonabant (1 mg/kg b.w., i.p.) and subsequent groups received the tested compounds (5 mg/kg b.w., i.p.). Rats were allowed to adapt for two days, they were given water in two identical bottles instead of one. Immediately following the first dose, and throughout the five-day injection period, the water in both bottles was replaced with freshly-made saccharin solution (1 g/L). This supply was withdrawn exactly 48 h before the last dosing (immediately before the 6th dose), and both bottles were again filled with water for another 24 h period. Exactly before the last dosing, rats were given access to one bottle filled with saccharin, and another filled with water. This two-bottle preference test was maintained for 24 h. During this time, saccharin intake, water intake, total fluid intake, the ratio saccharin/total fluid were registered. Body weight and the amount of consumed feed were also measured. The volume of saccharine solution consumed in relation to the total fluid intake was used as a measure of CTA, induced by treatment, and was expressed as a percentage of the saccharin intake. The scheme of the experiment is shown in Figure 7C.

### 4.7. Data Analysis and Statistical Procedures

Statistical calculations were carried out with the GraphPad Prism 6 program (GraphPad Software, San Diego, CA, USA). The results were given as arithmetic means with standard errors of the mean (SEM). The normality of data sets was determined using the Shapiro–Wilk test. The statistical significance was calculated using one-way ANOVA post-hoc Tukey’s Multiple Comparison Test or Multiple t tests (locomotor activity). Differences were considered statistically significant at: **p* ≤ 0.05, ***p* ≤ 0.01, ****p* ≤ 0.001.

## Figures and Tables

**Figure 1 pharmaceuticals-14-00270-f001:**
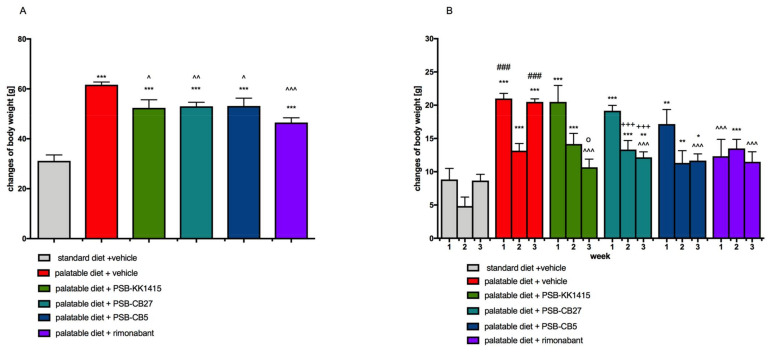
Effect of diet or long-term administration of the GPR18 ligands or rimonabant, on body weight of female Wistar rats in the model of excessive eating. (**A**) Sum of body weight changes. (**B**) Changes of body weight in individual weeks. Result are means ± SEM, *n* = 6. Multiple comparison against the vehicle-treated control group (*) or against the vehicle-treated obese control group (^) or against the vehicle-treated obese control group in second week (#) or against the PSB-KK1415-treated group in first week (o) or against the PSB-CB27-treated group in first week (+) were performed by one-way ANOVA Tukey post hoc. Significant differences are denoted by *,^,o *p* < 0.05; **,^^ *p* < 0.01; ***,^^^,###,+++ *p* < 0.001.

**Figure 2 pharmaceuticals-14-00270-f002:**
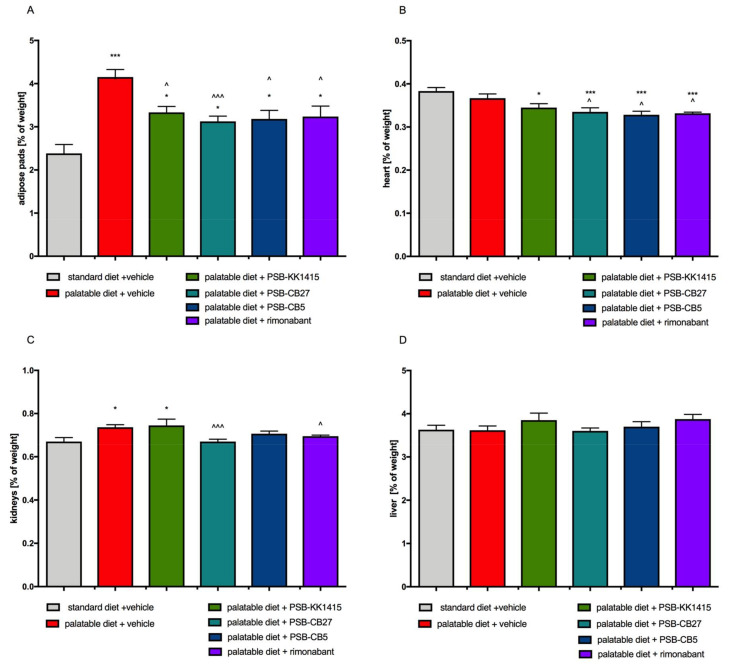
Effect of diet or long-term administration of the GPR18 ligands or rimonabant, on weights of various organs of female Wistar rats in the model of excessive eating. (**A**) Peritoneal adipose tissue. (**B**) Heart. (**C**) Kidneys. (**D**) Liver. Results are the means ± SEM, *n* = 6. Multiple comparisons against the vehicle-treated control group (*) or against the vehicle-treated obese control group (^) were performed by one-way ANOVA Tukey post hoc; Significant differences are denoted by *,^ *p* < 0.05; ***,^^^ *p* < 0.001.

**Figure 3 pharmaceuticals-14-00270-f003:**
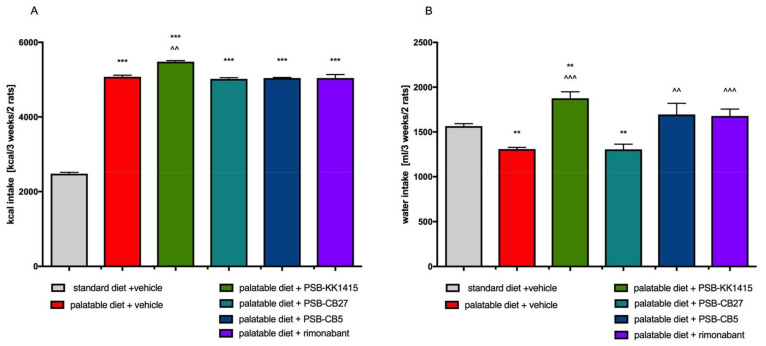
Effect of long-term administration of the GPR18 ligands or rimonabant, on food (**A**) or water intake (**B**) of female Wistar rats fed a palatable feed. Results are the means ± SEM, *n* = 6. Multiple comparisons against the vehicle-treated control group (*) or against the vehicle-treated obese control group (^) were performed by one-way ANOVA Tukey post hoc; Significant differences are denoted by **,^^ *p* < 0.01; ***,^^^ *p* < 0.001.

**Figure 4 pharmaceuticals-14-00270-f004:**
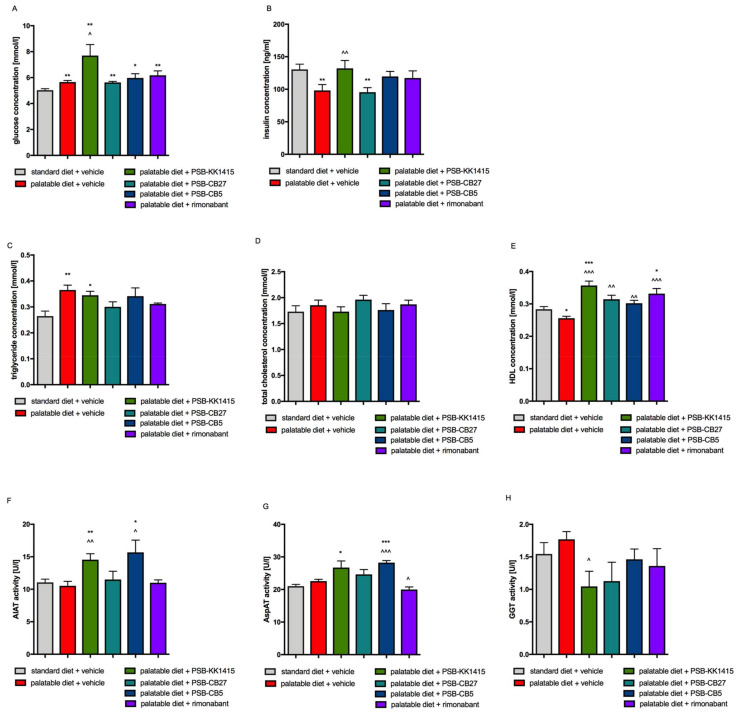
Effect of administration of the GPR18 ligands or rimonabant on plasma glucose (**A**), insulin (**B**), triglyceride (**C**), total cholesterol (**D**) or HDL-cholesterol (**E**) level or alanine aminotransferase (**F**), aspartate aminotransferase (**G**), γ-glutamyl transpeptidase, and (**H**) activity of female Wistar rats fed a palatable feed. Results are the means ± SEM, *n* = 6. Multiple comparisons against the vehicle-treated control group (*) or against the vehicle-treated obese control group (^) were performed by one-way ANOVA Tukey post hoc; Significant differences are denoted by *,^ *p* < 0.05, **,^^ *p* < 0.01, ***,^^^ *p* < 0.001.

**Figure 5 pharmaceuticals-14-00270-f005:**
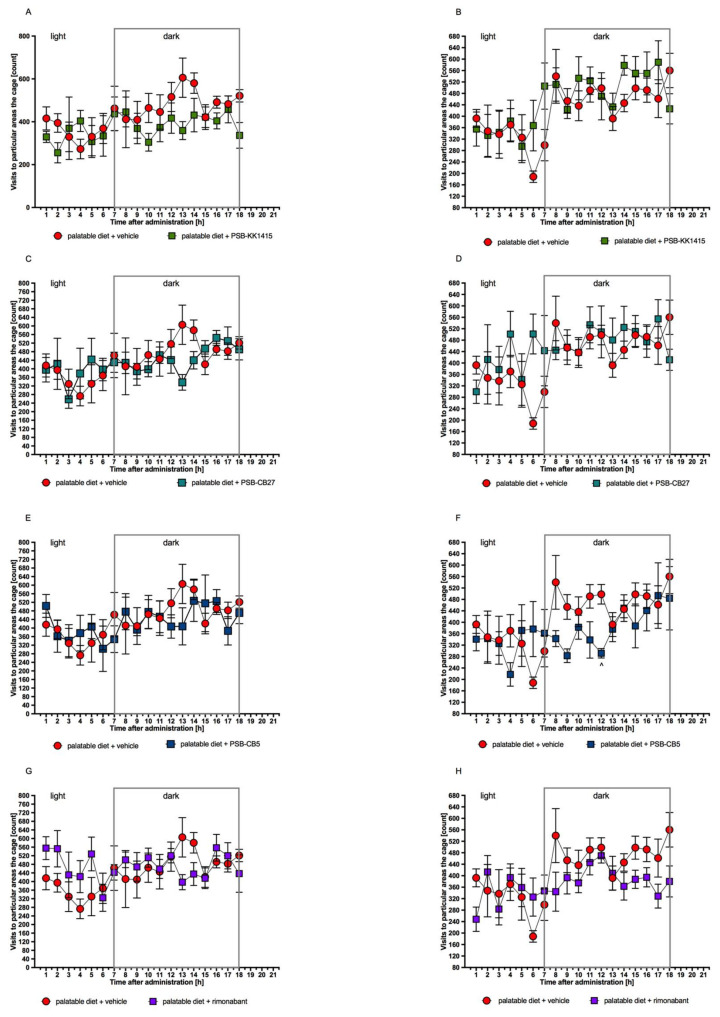
Influence of the GPR18 ligands or rimonabant on locomotor activity after a single dose (**A**,**C**,**E**,**G**) and chronic treatment (**B**,**D**,**F**,**H**). Locomotor activity of rats fed palatable feed during 18 h period after treatment with tested compounds (5 mg/kg b.w., i.p.), rimonabant (1 mg/kg b.w.), or vehicle. Activity is directly related to entrance to various areas of the cage. Results are the means ± SEM, *n* = 6 (multiple *t*-test). Significant differences are denoted by ^ *p* < 0.05.

**Figure 6 pharmaceuticals-14-00270-f006:**
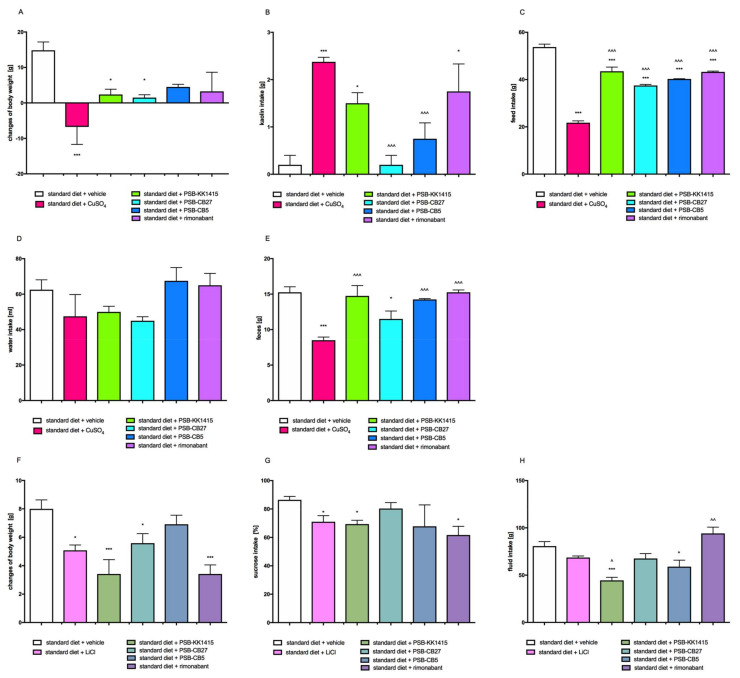
Effect of single administration of the GPR18 ligands or rimonabant, on body weight (**A**), kaolin intake (**B**), food intake (**C**), water intake (**D**), the amount of feces (**E**) of female Wistar rats in the Pica behavior model. The effect of multiple (seven days) administration of the GPR18 ligands or rimonabant, on body weight (**F**), sucrose intake (**G**), and fluid intake (**H**) of female Wistar rats in conditioned tested aversion model. Results are the means ± SEM, data for two animals are reared together, *n* = 6. Multiple comparisons against the vehicle-treated control group (*) or against the vehicle-treated obese control group (^) were performed by one-way ANOVA Tukey post hoc. Significant differences are denoted by *,^ *p* < 0.05; ^ *p* < 0.01; ***,^^^ *p* < 0.001.

**Figure 7 pharmaceuticals-14-00270-f007:**
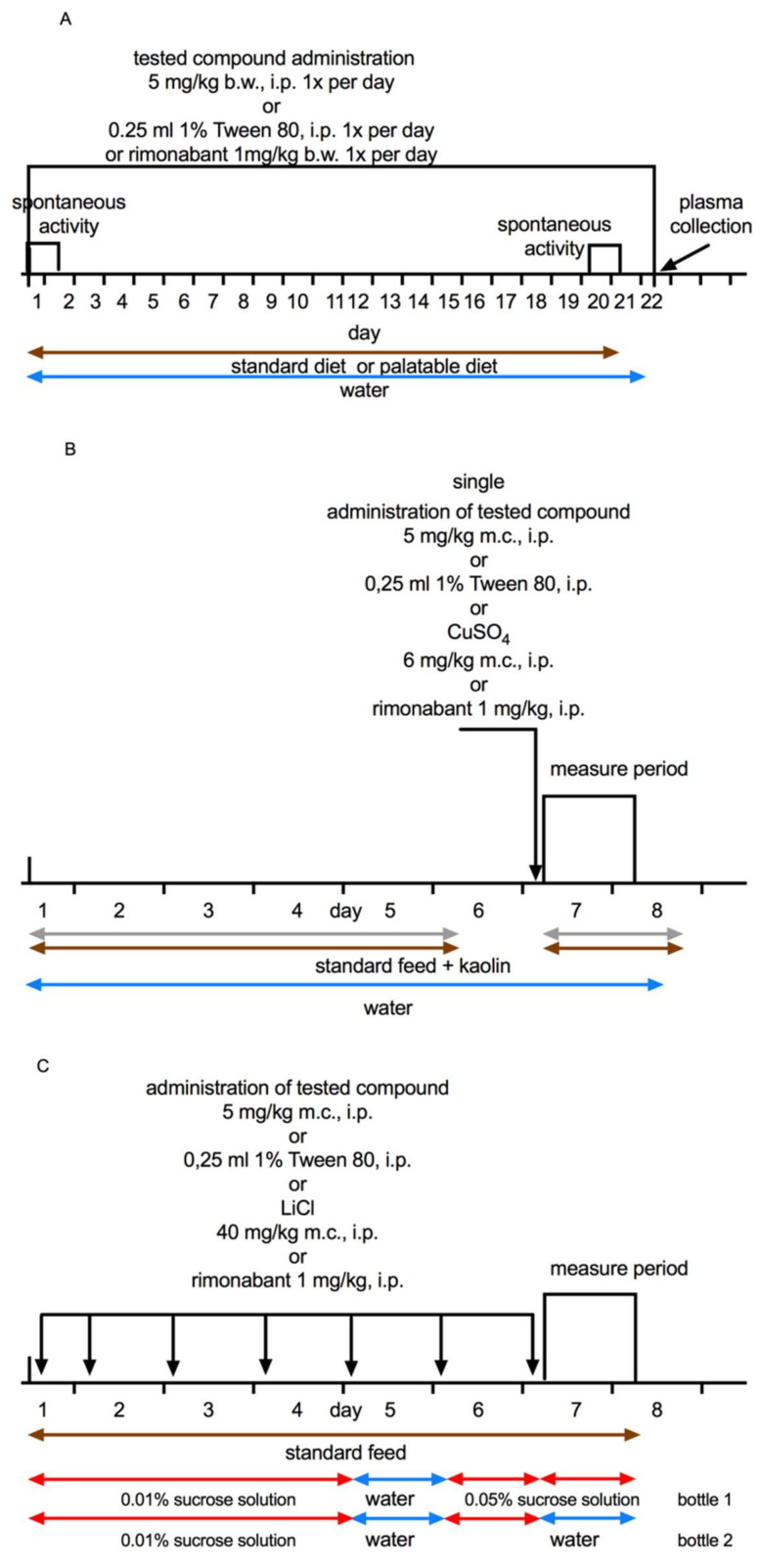
A schematic diagram of chronic administration of tested compounds in model of excessive eating. PSB-KK1415 or PSB-CB27 or PSB-CB5 (5 mg/kg b.w.) or rimonabant (1 mg/kg b.w.) were administrated intraperitoneally (i.p.) to rats for 22 consecutive days. Control groups received 1% Tween 80 (**A**). A schematic diagram of pica behavior test. Rats were treated with single dose of tested compounds (5 mg/kg b.w., i.p.) or rimonabant (1 mg/kg b.w., i.p.). Control group received 1% Tween 80 or CuSO4 (6 mg/kg b.w., i.p.) (**B**). A schematic diagram of conditioned tasted aversion test. Rats were treated with repeated (seven-times) dose of tested compounds (5 mg/kg b.w., i.p.) or rimonabant (1 mg/kg b.w., i.p.). Control group received 1% Tween 80 or LiCl (40 mg/kg b.w., i.p.). The measurement period began on the seventh day of administration and lasted for the next 24 h (**C**).

## Data Availability

The data presented in this study are available on request from the corresponding author.

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
