# Peer review of "Effects of GPR18 Ligands on Body Weight and Metabolic Parameters in a Female Rat Model of Excessive Eating"

_pharmaceuticals, 2021, doi:10.3390/ph14030270_

Round 1

Reviewer 1 Report

The submitted manuscript is a very interesting study on the metabolic effects of GPR18 ligands tested at different dietary conditions in female rats. Despite the high significance of the study, mandatory information are missing  and a  better organization of graphs/text is suggested.

Major points:

  • Introduction does not well describe the endocannabinoid system, with the appearance of GPR55 without any explanation: please, clearly indicate cannabinoid receptors and their ligands in the first part of the introduction.

  • This reviewer finds the presentation of results quite confusing with figures 1 and 2 first described focusing on control animals and then to treatments groups. In this respect, figure presentations that do not follow the explanation in the main text; this causes a continuous jumping from one figure to another one, a partial and mixed description of graphs as a whole, making the manuscript difficult to read.

  • Ethical statement for animal experimentation is lacking; similarly this reviewer does not find the authorization and approval (code) by Ethical Committee (mandatory item for studies in animal models).

  • 74-76 “The levels of glucose and triglyceride in blood were higher in obese rats than in the control group, but HDL-cholesterol level was lower (Figure 2A, C and E)” Also insulin levels were lower in obese rats than in the control group (fig 2B). However the discussion of glucose levels in parallel to insulin levels is described in par 2.3; lipid profile is described in par 2.4. Why such a duplication in the presentation of results?.

  • In each treatment group changes in body weight have to be presented comparing 1, 2 and 3 weeks each other. ThTR treatment seems time dependently to interfere in body weigh gain, for example. Why at 2 week body weight seems to decrease vs 1 week in control animal fed with palatable diet? Are these differences significant?

  • 134”However, overall levels of GGT activity were low” This sentence does not find confirmation in graph: missing statistics?

  • Presented data not fully sustain the improvement in lipid profile reported in the title.

Minor points:

  • English style requires revision

  • Why female rat only were used?

  • Figure 4: for consistency use different colours than previous figures for different dietary interventions.

  • Par 2.7 Insert references to the corresponding figure 4 panels

  • 317-327: there is something missing and/or confusing in the description of the used agonists/antagonists.

  • Elisa test lacks intra-/inter-coefficients of variability, detection limits,  replicates and  kit references

Reviewer 2 Report

The authors investigate the effects of a novel GPR18 ligands on obesity prevention. 

Comments:

As the authors only used female rats, this needs to be included in the title, and justified in the introduction.

although there are different reports on  this particular ligand-line 49, please clarify what this means?

Line 60- do these ligands also target GPR55? cite reference

(not yet published) should be replaced with unpublished observations

What is your hypothesis?

In the methods organ weights were recorded but not presented. These should be reported in the results.

Would the administration of 187 selective GPR18 ligands prevent induction of at least some pathological disorders? Could 188 these compounds act anorectically? Which ligand - agonist or antagonist would have the  most beneficial effect? This section should be rephrased to be statements not questions

statistics-were the data normally distributed and were any outliers removed?

I would replace selective antagonism to specific antagonism. Though not directly stated I am assuming that these compounds only target GPR18 and have poor selectivity for other receptors. 

Reviewer 3 Report

The study by Kieć-Kononowicz et al reported data concerning the effect of potential GRP18 antagonists (CB5, CB27) and agonist  (TH-Tr) on body mass and development of selected metabolic disorders commonly accompanying obesity. This is an interesting work which adds new toll to the current armamentarium of GPR18 ligands currently under investigation. However, in my opinion, some concerns need to addressed in order to fortify author conclusion.

  • why you used 5 mg/kg from TH-Tr; CB27; CB5? what is LD value for tested ligands?
  • With reference to the tested compounds, why the authors used expression excessive eating not obesity?
  • It would be interesting to show the specificity of ligands to bind to GPR18 using computational study. Recently the authors showed that they can perform a docking study (doi.org/10.3390/biom10050686) for GPR18. So the system is already established and can be done easily.
  • What gap in the current scientific landscape on GPR18 ligands is filled by this work?
  • why authors used in the paper title '' New GPR18 ligands'' although the ligands are known?
  • the authors claimed that the effects of GPR18 antagonists were specific. I think the authors should be careful in using the term of specificity, as it could be also GPR55. I would avoid it.
  • the introduction part is very short and lack from citing the relevant studies. The authors should discuss more about GPR18, obesity, GPR55..
  • in line 61-62, the authors mentioned that the availability of a selective GPR18 ligands may become a useful tool in research focused on the mechanisms of action of this receptor. What could be the difficulty in obtaining a selective antagonist for GPR18 receptor? please add something in this direction.
  • How the presented ligands by the authors are different from the rimonabant that are currently being investigated for obesity? it is clear from the results that the activity of rimonabant (1mg) is higher than that of the investigated ligands at 5mg?

minor points:

  • please reformat phrase 47-50
  • please improve the conclusion part

Round 2

Reviewer 1 Report

The authors have addressed most queries and the manuscript has been significantly improved. 

I only ask them 1) to consistently use the abbreviation AEA for anandamide

2)  Are they sure that in figure 6 the same colours have been used for the same treatments in the different panels? It seems they are all slightly different. 

Author Response

The authors have addressed most queries and the manuscript has been significantly improved. 

I only ask them 1) to consistently use the abbreviation AEA for anandamide

It has been corrected.

2)  Are they sure that in figure 6 the same colours have been used for the same treatments in the different panels? It seems they are all slightly different. 

The colors have been changed.

Reviewer 3 Report

Thanks to the authors for covering all comments that have been raised. I think now the MS has been extensively modified and I recommend the publishing of this study in the current form.

Author Response

Thank you.